# Distance, difference in altitude and socioeconomic determinants of utilisation of maternal and child health services in Ethiopia: a geographic and multilevel modelling analysis

Atkure Defar ,[1,2] Yemisrach B. Okwaraji,[1,3] Zemene Tigabu,[4] Lars Åke Persson [1,3] Kassahun Alemu[2]

For numbered affiliations see end of article.

**Correspondence to**
Dr Atkure Defar;
atkuredefar@gmail.com

## ABSTRACT

**Objective** We assessed whether geographic distance and difference in altitude between home to health facility and household socioeconomic status were associated with utilisation of maternal and child health services in rural Ethiopia.

**Design** Household and health facility surveys were conducted from December 2018 to February 2019.

**Setting** Forty-six districts in the Ethiopian regions: Amhara, Oromia, Tigray and Southern Nations, Nationalities, and Peoples.

**Participants** A total of 11 877 women aged 13–49 years and 5786 children aged 2–59 months were included.

**Outcome measures** The outcomes were four or more antenatal care visits, facility delivery, full child immunisation and utilisation of health services for sick children. A multilevel analysis was carried out with adjustments for potential confounding factors.

**Results** Overall, 39% (95% CI: 35 to 42) women had attended four or more antenatal care visits, and 55% (95% CI: 51 to 58) women delivered at health facilities. One in three (36%, 95% CI: 33 to 39) of children had received full immunisations and 35% (95% CI: 31 to 39) of sick children used health services. A long distance (adjusted OR (AOR)=0.57; 95% CI: 0.34 to 0.96) and larger difference in altitude (AOR=0.34; 95% CI: 0.19 to 0.59) were associated with fewer facility deliveries. Larger difference in altitude was associated with a lower proportion of antenatal care visits (AOR=0.46; 95% CI: 0.29 to 0.74). A higher wealth index was associated with a higher proportion of antenatal care visits (AOR=1.67; 95% CI: 1.02 to 2.75) and health facility deliveries (AOR=2.11; 95% CI: 2.11 to 6.48). There was no association between distance, difference in altitude or wealth index and children being fully immunised or seeking care when they were sick.

**Conclusion** Achieving universal access to maternal and child health services will require not only strategies to increase coverage but also targeted efforts to address the geographic and socioeconomic differentials in care utilisation, especially for maternal health.

**Trial registration number** ISRCTN12040912.

### Strengths and limitations of this study

- ► We assessed geographic and social equity across the continuum of maternal and child healthcare, that is, antenatal care visits, facility deliveries, immunisation and utilisation of health services for sick children.
- ► We used a multilevel regression modelling that included the determinants, the outcome and potential confounding factors at the cluster, household and individual levels accounting for confounding and clustering effects.
- ► We limited reporting of maternal services utilisation to the last 12 months and care seeking for common childhood illnesses to a 2-week recall period to reduce recall bias.
- ► In addition to the straight-line distance between home and the nearest health facility, we assessed the difference in altitude between home and the nearest health facility as a potentially important dimension of distance that could affect care utilisation.
- ► The distance measured between household and health facility does not entirely reflect the actual distance on the ground.

## INTRODUCTION

The global goal of universal health coverage is a significant challenge since half of the world's population does not have access to essential health services.[1] The Sustainable Development Goals stress the need for accelerating the ambitions to reach the targets for improving maternal, newborn and child health.[2] The WHO, with its partners, has developed policies and programmes aiming to promote health equity across and within nations.[3] Utilisation of maternal, newborn and child health services has increased. However, inequities by geographic location and socioeconomic status are challenges in the efforts towards universal health coverage.[4 5]

Studies from low-income African countries have indicated a strong association between geographic accessibility to health facilities and the use of maternal and child health services.[6] In Ethiopia, reports have described geographical inequality. Distance measured as a straight line and walking time was associated with attendance to antenatal care and facility delivery.[7–9] The topography of Ethiopia, characterised by difficult steep terrain with high hills, valleys and long distances from home to health facilities, has contributed to inequalities in the use of maternal and child health services.[10] In addition, studies in Ethiopia and other African countries have shown that distance to health facilities was negatively associated with the use of family planning,[11] antenatal care,[12] facility deliveries[13] and care seeking for sick children[14]

To the best of our knowledge, no previous studies have simultaneously analysed the difference in altitude, the distance between household and health facility as well as socioeconomic characteristics and their possible association with utilisation of a range of maternal and child health services in Ethiopia. Therefore, this study aimed to analyse if the geographic distance from home to the nearest health facility, the difference in altitude between home and health facility and socioeconomic status were associated with the use of antenatal care, facility delivery, full child immunisation and utilisation of primary care services for sick children in rural Ethiopia.

## METHODS
### Study design and setting
We conducted a cross-sectional study in 46 districts of the four most populous Ethiopian regions, namely, Amhara, Oromia, Tigray, and Southern Nations, Nationalities, and Peoples (figure 1). The regions have a population

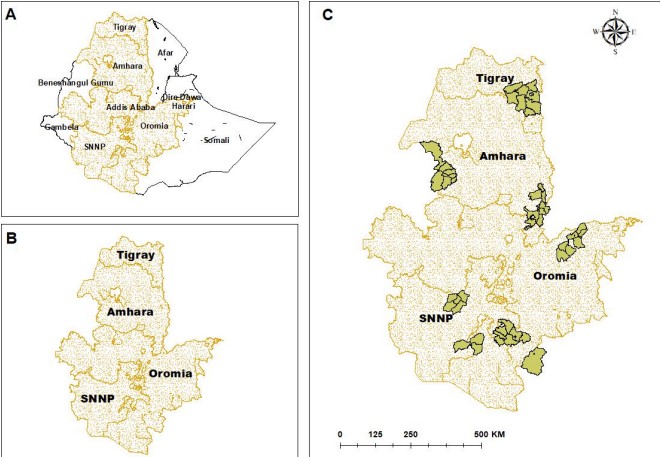

**Figure 1** Maps showing (A) Ethiopia, (B) the four study regions (Tigray, Amhara, Oromia, SNNP (South Nation, Nationalities and People) and (C) the 46 study districts in the four regions (graph constructed in ArcGIS V.10.4 software using freely available data from the Ethiopian Central Statistics Agency).

scattered in dispersed settlements, and the primary source of income is farming.

In the Ethiopian health delivery system, the primary healthcare unit is comprised of five satellite health posts and one health centre. Each health post serves approximately 5000 people and is staffed by two female health extension workers.

The study was conducted from December 2018 to February 2019. The Ethiopian Public Health Institute, in collaboration with the London School of Hygiene and Tropical Medicine, University of Gondar, Mekelle University, Jimma University and Hawassa University, conducted the survey.

### Sampling
This study used data from an evaluation of the Optimizing of Health Extension Program intervention that was implemented in four regions of Ethiopia. This intervention aimed to improve care utilisation for sick under 5 children.[15] The evaluation included baseline and endline surveys, and this study was a secondary analysis of end-line data.

A two-stage stratified cluster sampling was applied in the selected study districts. The first stage used lists of enumeration areas from the 2007 Ethiopian Housing and Population Census as the sampling frame.[16] Thereafter, 194 enumeration areas were selected with probability proportional to size (sampling from a finite population in which a size measure is available for each population unit before sampling and where the probability of selecting a unit is proportional to its size).

In the second stage, all households within each enumeration area were listed and a sampling interval was calculated. A random start number between 1 and the sampling interval was selected. The households that matched the random start number on the list were chosen as the first household to be included. This process was repeated until the targeted number of 60 households in each cluster was reached. All women aged 13–49 years and children under the age of 5 years, who lived in the selected households, were included in the study. The health posts and the health centres serving the enumeration area were included in the study.

We used a standard sample size formula to calculate the sample size. We estimated a design effect of 1.3, 80% power and the assumption of a ratio of children less than 5 years of age per household was considered, based on sampling reported in the Ethiopia Demographic Health Survey 2011.[17] We used the assumption to detect 15% improved childcare utilisation in two groups (intervention and comparison) between baseline and end-line surveys. The sample size was estimated to be 6000 households per group (12 000 in total) to have 80% power to detect differences of 15 percentage points for care seeking for 2–59 months children. The protocol for the evaluation study was registered with a trial number ISRCTN12040912 and published.[15]

## Data collection tools

We used a modular questionnaire that was based on the demographic and health survey and similar assessment tools. Information on the antenatal care attendance and place of delivery was collected from all women aged 13–49 who had a live birth in the last 12 months before the survey. Caregivers of children aged 2–59 months were asked whether the child had any illness in the previous 2 weeks and if they had sought care from an appropriate provider. Caregivers were invited to show the immunisation cards and asked additional questions on the different vaccinations if the card was not available. Besides, the questionnaire included information on sociodemographic data, assets and geographic coordinates of households and health facilities. The coordinates were measured by geographical positioning system (GPS) dongles (ND-100S).

## Definition of variables
### Outcome variables

We estimated the overall coverage of four selected maternal and child health services. For women, this included (1) the proportion of women aged 13–49 with a live birth within the last 12 months before the survey who had attended antenatal care four or more times and (2) the proportion of women with a live birth within 12 months before the survey who gave birth in a health facility. For the child health services utilisation, we included: (1) the proportion of children aged 12–23 months who had received full immunisation, which was defined as BCG, three pentavalent vaccinations (diphtheria, tetanus, pertussis, hepatitis B and *Haemophilus influenzae*), oral polio vaccine and one dose of measles vaccine[18] and (2) the proportion of children aged 2–59 months with fever, diarrhoea or suspected pneumonia (cough and difficult or fast breathing) in the last 2 weeks for whom care was sought from an appropriate provider, that is, health posts, health centres, hospitals or private clinics.

### Explanatory variables

We used the Euclidian distance and the difference in altitude from the participants' houses to the nearest health facility and household wealth as explanatory variables. Furthermore, we included other sociodemographic variables such as age, sex, parity, region, caregiver's education level and number of children in the household to adjust for potential confounding in the analyses.

## Data management and analysis

Household data on care utilisation were linked with the GPS information of the 142 health centres and 164 health posts that served the households. The linking was done using cluster identification. The Euclidian distance from participants' houses to the nearest health facility was calculated using the geographical location of health facilities and households. Thus, we calculated the distance to the nearest health post for child health services (full child immunisation and sick child healthcare utilisation) and distance to the nearest health centre for maternal healthcare services (antenatal care and facility delivery). These distances were divided into tertiles and labelled as short, medium and long. Similarly, the difference in altitude was estimated using the altitude information of both the households and the health facility and classified as small, medium and large. Corrections of the GPS readings were done whenever the reading erroneously was positioned outside the study areas. If that was the case, we adjusted by considering the nearest GPS reading captured. To estimate household wealth, we constructed wealth index using 15 household assets and characteristics. We used principal component analysis to construct the wealth index and divided it into tertiles.[19]

We first analysed the prevalence of the outcomes variables and graphically displayed the relation between distance, the difference in altitude, household wealth and facility visits for antennal care, facility delivery, child immunisation and sick child services utilisation. After that, we used two-level mixed-effect logistic regression modelling to examine the associations with individual-level variables (wealth, number of children in the household, sex, age, parity and educational status), group-level variables (region, distance and the difference in altitude) and the interactions between the two. We analysed the crude and adjusted ORs with 95% CIs. As individual observations were grouped within the cluster specifically by geography (region, distance to the health facility and their altitude), the regression models we used accounted for confounding and clustering effects. Considering both the distance as well as the difference in altitude would yield a good estimate of the extent of space between the health facilities and the households. The model quality was assessed using Akaike's information criterion and intracluster correlation coefficient. The significance level was set at a p value of less than 5%. All analyses were done using STATA V.15 (Stata Corporation, College Station, Texas).

## Patient and public involvement

Patients or the public were not involved in the design or conduct, or reporting, or dissemination plans of this research.

## RESULTS
### Characteristics of the study participants

Out of 11 877 women aged 13–49, 957 had given birth during 12 months before the survey. Of these, we linked 766 women (80%) with the geographic location data of the 142 nearest health centres, since antenatal and delivery services were provided at the health centre level or above (table 1). We collected information on 5786 children from 2 to 59 months of age. Of these, we linked data on 5285 (91%) children with the geographic location of 164 nearest health posts. We estimated the Euclidian distance (median: 9.7, IQR: 5.2–14.4 km) and the difference in

**Table 1** Number of women and children with data linked to the health facility geographical positioning system information that was used to calculate distance and difference in altitude between home and health facility

| Group | Number of cases | Number of health facilities linked* | Cases linked† |
|---|---|---|---|
| Children: 2–59 months | 5787 | 164 health posts | 5285 (91%) |
| Children: 12–23 months | 992 | 164 health posts | 912 (92%) |
| Women: 13–49 years | 957 | 142 health centres | 766 (80%) |

*Number of health facilities assessed in the study area that were supposed to provide service to the study population.
†Number of study participants whose data were linked to the health facility location to calculate the distance and difference in altitude between the household and the health facility.

altitude (median: 54, IQR: 24–1312 m from the participants' houses to the nearest health facility.

### Utilisation of maternal health services
Of the 766 mothers who had given birth in the previous 12 months before the survey, 39% (95% CI: 35 to 42) had attended antenatal care four or more times. The proportion of women who had delivered at a health facility was 55% (95% CI: 51 to 58). There was an inverse relationship between distance, the difference in attitude and antenatal care or facility delivery (figure 2). There was an increased proportion of prenatal visits and the percentage of facility deliveries with an increasing wealth index.

### Utilisation of child health services
Among the 912 children in the age group 12–23 months, 36% (95% CI: 33 to 39) were fully immunised. The 2-week morbidity with common childhood illnesses, that is, fever, diarrhoea or suspected pneumonia (cough with fast or difficult breathing), was assessed among 5285 children from 2 to 59 months of age. Of these, 9.4% (95% CI: 8.6 to 10.2) were found to have been sick. The proportion of children who sought care from an appropriate provider, such as health post, health centre, hospital or private clinics, was 35% (95% CI: 31 to 39). There was no clear pattern in the associations between distance, the difference in altitude, wealth index and child immunisations or utilisation of services for sick children (figure 2).

### Determinants of maternal and child health services utilisation
The multilevel regression analysis, after adjusting for age, parity, education and region, showed that women living with higher difference in altitude were less likely to have attended antenatal care four or more times (adjusted OR (AOR)=0.46; 95% CI: 0.29 to 0.74) and to have delivered at a health facility (AOR=0.33; 95% CI: 0.19 to 0.59) table 2. Women living at a long distance from health facilities were less likely to have delivered at health facility (AOR=0.57; 95% CI: 0.34 to 0.96). Women from the wealthier household were more likely to have received antenatal care four or more times (AOR=1.67; 95% CI: 1.02 to 2.75) and delivered at a health facility (AOR=3.70; 95% CI: 2.11 to 6.47) than their counterparts in the lower wealth tertiles (table 2).

There was no clear pattern in the associations between distance from home to health post, the difference in altitude or household wealth and child immunisation and sick childcare utilisation (table 3). There was no indication of socioeconomic inequity in coverage of immunisation or utilisation of services for the sick child. The medium distance stratum showed a lower coverage.

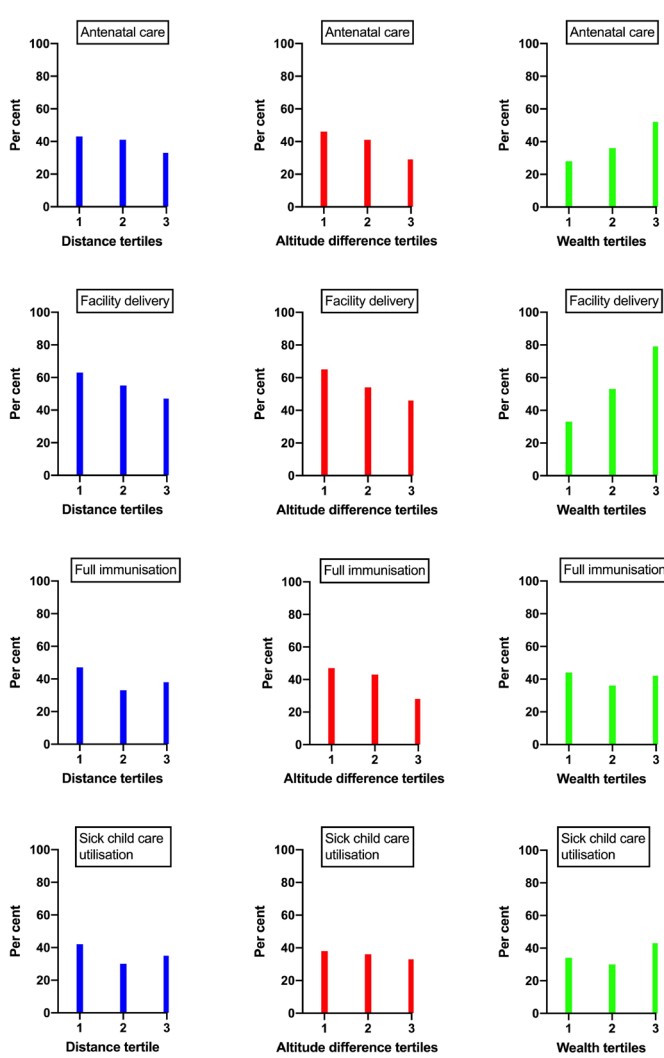

**Figure 2** The distribution of antenatal care, facility delivery, child immunisation and sick childcare utilisation by tertiles of the distance and difference in altitude between the household and the nearest health facility and the wealth index.

**Table 2** Determinants of antenatal care four or more times and facility delivery in four regions of Ethiopia, 2019

| Characteristic | Category | Antenatal care four or more visits | | Institutional delivery | |
|---|---|---|---|---|---|
| | | OR (95% CI) | AOR (95% CI) | OR (95% CI) | AOR (95% CI) |
| Household and individual-level characteristics | | | | | |
| Wealth tertiles | Low | Ref. | Ref. | Ref. | Ref. |
| | Middle | 1.40 (0.90 to 2.19) | 1.15 (0.74 to 1.81) | 2.48 (1.54 to 4.01) | 1.61 (0.99 to 2.60) |
| | Upper | 2.50 (1.55 to 4.02) | 1.67 (1.02 to 2.75) | 6.85 (3.87 to 12.15) | 3.70 (2.11 to 6.48) |
| Age | <20 | Ref. | Ref. | Ref. | Ref. |
| | 21–25 | 1.84 (1.01 to 3.37) | 1.52 (0.80 to 2.89) | 0.62 (0.34 to 1.17) | 0.72 (0.36 to 1.43) |
| | 26–30 | 2.33 (1.29 to 4.22) | 1.74 (0.85 to 3.53) | 0.71 (0.38 to 1.32) | 0.99 (0.45 to 2.15) |
| | 31–35 | 1.99 (1.02 to 3.91) | 1.51 (0.68 to 3.40) | 0.85 (0.41 to 1.76) | 1.35 (0.54 to 3.34 |
| | 36–49 | 1.97 (0.90 to 4.28) | 1.42 (0.56 to 3.61) | 0.78 (0.33 to 1.85) | 1.34 (0.47 to 3.82) |
| Educational level | No school | Ref. | Ref. | Ref. | Ref. |
| | Schooling | 1.12 (0.79 to 1.60) | 1.18 (0.80 to 1.74) | 1.53 (1.02 to 2.29) | 1.26 (0.81 to 1.96) |
| Parity | One | Ref. | Ref. | Ref. | Ref. |
| | Two | 0.92 (0.51 to 1.68) | 1.01 (0.55 to 1.87) | 0.69 (0.34 to 1.41) | 0.94 (0.46 to 1.93) |
| | Three | 0.69 (0.37 to 1.30) | 0.77 (0.39 to 1.52) | 0.37 (0.18 to 0.74) | 0.48 (0.22 to 1.05) |
| | Four | 1.35 (0.74 to 2.47) | 1.44 (0.73 to 2.84) | 0.41 (0.20 to 0.84) | 0.62 (0.28 to 1.40) |
| | ≥Five | 0.96 (0.56 to 1.62) | 1.01 (0.56 to 2.15) | 0.38 (0.20 to 0.71) | 0.47 (0.21 to 1.03) |
| Group-level characteristics | | | | | |
| Region | Tigray | Ref. | Ref. | Ref. | Ref. |
| | Amhara | 0.93 (0.49 to 1.70) | 0.75 (0.41 to 1.36) | 1.99 (0.91 to 4.39) | 1.35 (0.66 to 2.77) |
| | Oromia | 0.23 (0.13.to 0.45) | 0.22 (0.12 to 0.41) | 0.13 (0.06 to 0.28) | 0.13 (0.06 to 0.26) |
| | SNNP | 1.48 (0.66 to 3.35) | 1.05 (0.49 to 3.34) | 2.32 (0.80 to 6.75) | 1.26 (0.48 to 3.34) |
| Distance to health facility tertiles | Short | Ref. | Ref. | Ref. | Ref. |
| | Medium | 1.06 (0.69 to 1.64) | 1.01 (0.67 to 1.54) | 0.87 (0.52 to 1.44) | 0.90 (0.56 to 1.47) |
| | Long | 0.75 (0.46 to 1.22) | 0.77 (0.49 to 1.19) | 0.51 (0.29 to 0.90) | 0.57 (0.34 to 0.96) |
| Difference in altitude tertiles* | Small | Ref. | Ref. | Ref. | Ref. |
| | Medium | 0.78 (0.49 to 1.22) | 0.85 (0.55 to 1.29) | 0.59 (0.34 to 1.02) | 0.60 (0.36 to 1.01) |
| | Great | 0.48 (0.28 to 0.83) | 0.46 (0.29 to 0.74) | 0.39 (0.19 to 0.79) | 0.34 (0.19 to 0.59) |
| ICC for the null model (%) | | | 27% | | 49% |
| ICC for the final model (%) | | | 8% | | 13% |

*Difference in altitude between household and nearest health centre.
AOR, adjusted OR; ICC, intraclass correlation coefficient.

## DISCUSSION

We have shown that long distance to the health centre, a more significant difference in altitude between home and facility and a lower socioeconomic group were all associated with a smaller proportion of women delivering at a health facility. Similarly, those living with a considerable difference in altitude and belonging to a lower socioeconomic group were less likely to have attended antenatal care four times or more. However, distance, altitude or socioeconomic status were neither associated with coverage of child immunisation nor with care seeking for sick children.

An estimated, 39% of the pregnant women in the current study had attended antenatal care four or more times. This proportion was higher than the corresponding result reported from the same study areas 2 years earlier, which showed coverage at 30%.[10] It was a bit lower than the national average, based on the Demographic and Health Survey 2019, which was 43%.[20] This increase could reflect the government's efforts towards universal health coverage.[21] In contrast to the current study, in an analysis of the 2016 Ethiopian Demographic and Health Survey, distance to the health centre was associated with four or more antenatal care visits.[7] Furthermore, in an analysis of data from Demographic and Health Surveys in five East African counties 2010–2014 showed that geographical accessibility of health facilities had a strong influence on maternal healthcare utilisation.[22] The study employed

**Table 3** Determinants of child immunisation and sick child services utilisation in four regions of Ethiopia, 2019

| Variable | Category | Full immunisation 12–23 months | | Sick child care utilisation 2–59 months | |
|---|---|---|---|---|---|
| | | OR (95% CI) | AOR (95% CI) | OR (95% CI) | AOR (95% CI) |
| Household and individual-level characteristics | | | | | |
| Wealth tertiles | Low | Ref. | Ref. | Ref. | Ref. |
| | Middle | 0.91 (0.52 to 1.62) | 0.75 (0.42 to 1.34) | 0.73 (0.39 to 1.37) | 0.71 (0.38 to 1.33) |
| | Upper | 1.58 (0.91 to 2.77) | 1.15 (0.61 to 2.17) | 1.02 (0.55 to 1.91) | 0.89 (0.42 to 1.88) |
| Sex | Girl | Ref. | Ref. | Ref. | Ref. |
| | Boy | 1.15 (0.84 to 1.57) | 1.18 (0.85 to 1.63) | 1.11 (0.73 to 1.66) | 1.02 (0.67 to 1.55) |
| Number of under 5 children in the household | One child | Ref. | Ref. | Ref. | Ref. |
| | ≥Two children | 1.10 (0.79 to 1.54) | 1.21 (0.86 to 1.72) | 1.15 (0.74 to 1.80) | 1.33 (0.83 to 2.14) |
| Group-level characteristics | | | | | |
| Region | Tigray | Ref. | Ref. | Ref. | Ref. |
| | Amhara | 1.32 (0.72 to 2.42) | 1.23 (0.62 to 2.45) | 2.07 (1.03 to 4.13) | 2.26 (1.02 to 5.03) |
| | Oromia | 0.49 (0.25 to 0.94) | 0.49 (0.25 to 0.95) | 2.87 (1.42 to 5.86) | 3.09 (1.54 to 6.19) |
| | SNNP | 0.68 (0.30 to 1.52) | 0.55 (0.23 to 1.34) | 2.04 (0.90 to 4.62) | 2.55 (1.07 to 6.09) |
| Distance to health facility tertiles | Short | Ref. | Ref. | Ref. | Ref. |
| | Medium | 0.69 (0.46 to 1.04) | 0.64 (0.41 to 0.98) | 0.49 (0.29 to 0.82) | 0.50 (0.29 to 0.85) |
| | Long | 0.84 (0.55 to 1.28) | 0.82 (0.53 to 1.27) | 0.76 (0.44 to 1.29) | 0.76 (0.44 to 1.29) |
| Difference in altitude tertiles* | Small | Ref. | Ref. | Ref. | Ref. |
| | Medium | 1.00 (0.66 to 1.47) | 0.99 (0.65 to 1.51) | 0.88 (0.54 to 1.44) | 1.02 (0.61 to 1.70) |
| | Great | 0.85 (0.54 to 1.35) | 0.89 (0.55 to 1.42) | 0.48 (0.28 to 0.82) | 0.59 (0.34 to 1.01) |
| ICC for the null model (%) | | | 24% | | 18% |
| ICC for the final model (%) | | | 19% | | 8% |

*Difference in altitude between household and nearest health facility.
AOR, adjusted OR; ICC, intraclass correlation coefficient.

cost–distance analysis based on the health facilities' position and difficulties in traversing the surface of each 300×300 m cell in the study areas. The analysis included the slopes based on elevation measurements as well as the distance from home to the health facility. These findings imply that efforts climbing up and down the hills were barriers to accessing these services. A similar result was shown regarding the influence of household wealth on antenatal care use and facility delivery in the previous study in the same study area.[19]

We found that more than half of the study women had delivered at a health facility, which was similar to the coverage (48%) reported by the 2019 Ethiopian Demographic Health Survey.[20] In line with the current study, there was also a marked wealth-based inequity in the utilisation of antenatal care four or more times and facility delivery in a study performed in the same areas 2 years earlier.[19] Reviews of similar Ethiopian studies,[23] research from sub-Saharan Africa countries[24] and East African countries[22] as well as from a range of low-income countries[25] point at considerable regional variation, social inequities in the utilisation of facility delivery services and the importance of topography and distance from home

to health centres and hospitals. A study from Kenya which attempted to disentangle distance to facility and quality of care indicated that pregnant women bypassed possible places to deliver, preferring to give birth at a facility that provided better-quality care.[26] Furthermore, an analysis from 81 low-income and middle-income countries suggested that closing the quality gap would enhance the use of services.[27] Such evidence suggest that the problem with distance to the health facility may be less important if the services offered maintain a high quality. In some areas, it may be motivated to consider outreach antenatal clinics organised by the health centres. The problems with distance to facilities for delivery services may be managed by schemes to organise transport or manage associated costs or by improving maternal waiting homes.

In terms of utilisation of child health services, 36% of children aged 12–23 months had received all basic immunisations. This low level was similar to the national estimate (38%) from 2019 Ethiopian Mini Demographic and Health Survey.[18] A study 2 years earlier in the same rural areas as the current study showed an equitable distribution of full child immunisation across different socioeconomic groups.[19] This relative equity in vaccination coverage

could be explained by the rural outreach services, organised from the health posts, where eligible children were summoned to vaccination days in their local villages.[19] Our analysis showed that the medium tertile of distance to health post was linked to lower immunisation coverage and sick child healthcare utilisation. This may be a random error, given the lack of consistency across distance strata. If true, it may be a reflection of the geographic distribution of outreach services. A study in Ethiopia's hard-to-reach areas, distance to the health facility influenced full child immunisation coverage in addition to the mother's education, utilisation antenatal delivery services and child delivered at the health facility.[28] However, in an analysis of the 2016 Demographic and Health Survey, full child immunisation showed significant variation between groups characterised by geographic, socioeconomic and maternal factors with proadvantaged bias.[29] The immunisations are given at the health post and partly through outreach services in the villages. This combination of facility-based and outreach services may theoretically not only increase coverage but also enhance equity.[30 31]

One-third of the children 2–59 months, who had been sick in the 2 weeks before the survey, had sought care from health posts, health centres, hospitals or private clinics, which is in line with the national estimates of the utilisation of health services for sick children.[32] Our results showed no association between care seeking for sick children and distance to the nearest health centre, difference in altitude and socioeconomic status. During the last decade, Ethiopia has scaled up the integrated community case management of common childhood illnesses provided by salaried health extension workers at the health posts.[33] The availability of community-level primary care services at health posts close to home, offering child health services, could partly explain the relative equity in utilisation. Contrary to this, the 2016 nationally representative Demographic Health Survey showed inequities in care seeking for common childhood illnesses between groups defined by geographic region, mother's education or household wealth.[32] The Ethiopian government has expanded maternal, neonatal and child health services in the country.[34 35] Further efforts are needed not only to expand these services but also to increase the quality of care[36 37] to reach the goal of universal health coverage.[21]

The difference in the influence of distance and difference in altitude between maternal and child health service may be related to the differences in service provision. The child health services, mainly managed by the health posts, include outreach activities with immunisation and home-based care. Women had to reach the more distant health centres to receive antenatal and delivery care.

The sample for this study was drawn from intervention and comparison areas for the evaluation of the Optimizing of Health Extension Program intervention, which was implemented in selected districts of the four most populous Ethiopian regions.[15] Thus, the results may not be generalisable to the population at country level.

We have, however, reasons to believe that the selected districts are typical for agrarian areas of these major Ethiopian regions. Previously published studies have focused on one or a few components of maternal and child care utilisation,[38 39] but we assessed the major service coverages, that is, antenatal care visits, facility delivery, immunisation and utilisation of health services for sick children. In cross-sectional studies like this, there is a risk of biased reporting of care utilisation. To reduce that risk, we limited the maternal services indicators to the last 12 months. The immunisation coverage adhered to the methodology of the demographic and health surveys, including children aged 12–23 months. Similarly, we followed the common practice of limiting common childhood illnesses and care seeking to a 2-week recall.

We assessed the straight-line distance between home and the nearest health facility. In reality, it may not be possible to travel in a straight line to a specific location and the route from home to the nearest health facility may be longer than the straight line as one has to avoid obstacles such as a river or a steep slope. In such cases, using cost–distance measurements had given a more realistic measurement. Some home-to-facility distance data were missing, especially regarding the maternal indicators, that is, the distance from home to the health centre. The reason was missing GPS coordinates of the health centre the family was supposed to use. We have no reason to believe that these omissions introduced any biased estimates. We added the difference in altitude as a potentially important dimension in care utilisation. In Ethiopia, with its mountains and valleys, the difference in altitude expresses some of the time and exact distance by road or path from home to the health facility.

We assessed altitude by GPS data that may not always be accurate. However, the error is usually of little importance, in the range from 10 to 20 m. The socioeconomic status was based on assets and characteristics of the household in the way that is practised in the demographic and health surveys. This characterisation represents accumulated wealth and resources but does not directly represent the economic strength of the household to cover costs for care seeking. Given the relative imprecision of the included determinants, we categorised these data into tertiles.

We used multilevel regression modelling that included the determinants, the outcome and potential confounding factors at the cluster, household and individual levels. The regression models accounted for confounding and clustering effects.

## CONCLUSION

The topography, that is, the distance and difference in altitude of the areas and socioeconomic status of the household affected the utilisation of maternal health services. We did not find such inequities in the utilisation of child health services, although coverage was on a low level. The strive for universal health coverage in Ethiopia

will require not only increased coverage but also target efforts to address the geographic and social differentials in care utilisation.

**Author affiliations**
[1]Health System and Reproductive Health, Ethiopian Public Health Institute, Addis Ababa, Ethiopia
[2]Institute of Public Health, Department of Epidemiology and Biostatistics, College of Medicine and Health Sciences, University of Gondar, Gondar, Ethiopia
[3]Department of Disease Control, London School of Hygiene and Tropical Medicine Faculty of Infectious and Tropical Diseases, London, UK
[4]Department of Paediatrics and Child Health, College of Medicine and Health Sciences, University of Gondar, Gondar, Ethiopia

**Acknowledgements** The authors would like to thank the University of Gondar for the support provided to the project. The authors are also grateful to the regional health bureaus and district health offices in the study areas for their smooth and kind cooperation. The authors would like to specially thank all field staff for their excellent fieldwork and the study participants for their kind cooperation in interviews and data collection.

**Contributors** The study was conceived by AD with inputs from LP and KA. Authors AD, LP, KA, ZT and YBO contributed in the acquisition, analysis or interpretation of data. AD prepared the first draft of the manuscript. LP, YBO, ZT and KA contributed to the interpretation of results and revision of the manuscript. All authors read and commented on the manuscript and approved the final version of the paper.

**Funding** The study was funded by a grant to the London School of Hygiene & Tropical Medicine from the Bill and Melinda Gates Foundation (OPP1132551).

**Map disclaimer** The depiction of boundaries on this map does not imply the expression of any opinion whatsoever on the part of BMJ (or any member of its group) concerning the legal status of any country, territory, jurisdiction or area or of its authorities. This map is provided without any warranty of any kind, either express or implied.

**Competing interests** None declared.

**Patient and public involvement** Patients and/or the public were not involved in the design, or conduct, or reporting, or dissemination plans of this research.

**Patient consent for publication** Not required.

**Ethics approval** Ethical approval from the London School of Hygiene & Tropical Medicine, London, UK (Ethics Ref 16117), the Ethiopian Public Health Institute (Ethics Ref 613/52) and the University of Gondar (O/V/P/RCS/05/214/2018) was obtained. During the data collection, written informed consent was obtained from household heads and women and assent from minors (age 13–18 years).

**Provenance and peer review** Not commissioned; externally peer reviewed.

**Data availability statement** Data could be available at a reasonable request to the corresponding author, AD, atkuredefar@gmail.com.

**ORCID iDs**
Atkure Defar http://orcid.org/0000-0001-9435-2135
Lars Åke Persson http://orcid.org/0000-0003-0710-7954

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
