## [Reviewer comments · BMJ Open]

ARTICLE DETAILS

TITLE (PROVISIONAL)	Distance, difference in altitude, and socio-economic determinants of utilization of maternal and child health services in Ethiopia: a geographic and multilevel modelling analysis
AUTHORS	Defar, Atkure; Okwaraji, Yemisrach; Tigabu, Zemene; Persson, Lars; Alemu, Kassahun

VERSION 1 – REVIEW

REVIEWER	Hui Luan University of Oregon
REVIEW RETURNED	28-Jul-2020

GENERAL COMMENTS	This paper uses a multilevel regression approach to explore distance, difference in altitude, and socioeconomic variables' impacts on maternal and child healthcare facility utilization in 46 districts of four Ethiopia regions. The topic is of significance and the authors did a fairly good job in describing the research background, strengths and limitations, and methodology. I have several comments before the paper can be accepted for publication. 1. The authors stressed that no previous studies included both distance between household and healthcare facility, and differences in altitude in the same study. It's best if the authors can provide more details in terms of why including both are important, probably from either the methodological or theoretical perspective, or both.2. How did the authors account for the clustering effects? More details are needed.3. I had difficulty understanding the characterization of the group-level variables used in the analysis, in particular the distance and the difference in altitude. If the distance/difference is between each household and the regional center or facility locations, why it is group-level rather than individual-level?4. In Table 3, the authors described the negative association between medium distance stratum and immunization/healthcare utilization as a result of random error. More clarifications are needed given that this finding exists for 3 out of the 4 models.5. Since the effects of distance to healthcare facilities on adult women and children were different based on Tables 2 and 3, what are the potential policy implications regarding improving women's and children's access to healthcare facilities? Would they be different? I'd like to see more discussions on this.6. A related question to Q5 is: while using the straight-line distance between home and the nearest healthcare facility to characterize healthcare access is controversial in the literature, are there any other measures the authors would consider to improve healthcare access characterization in addition to network distance-based measures?
---

REVIEWER	Gilbert Abiuro University for Development Studies, Ghana
REVIEW RETURNED	14-Oct-2020

GENERAL COMMENTS	Abstract - The abstract covers the essential elements of an abstract, however, the following should be considered.  1. Page 3, line 10, replace “socio-economic group” with “socio-economic status” 2. Page 3, line 14, replace “was” with “were” in the statement “A household and health facility surveys was conducted...” 3. What is the rationale for the 95% confidence intervals around the percentages? This comment applies to the results section as well Introduction 4. Page 6, line 3, replace “stresses” with “stress” 5. Page 6, line 10, replace “socio-economic group” with “socio-economic status” Methods  6. Page 8 line 31, replace “was using” with “used” 7. Page 8, line 45. The statement “194 enumeration areas were selected with probability proportional to size” is not clear. Please expatiate 8. Page 10, line 22-23. Was the nearest facility the same as the facility that was used for the various maternal and child health services under consideration? How was the situation whereby, the nearest facility was bypassed for a distant facility treated in the study? 9. Page 11, line 33. The statement “patients and the public were not involved in this study” is not clear. From whom was the data collected? Results  10. Page 12, line 10 “table 2” is hanging. What is about table Discussion  11. Page 1-13. I am not clear with the connection of the Kenya study on bypassing with the results of the current study. General comment: The entire manuscripts must be edited to correct a number of grammatical and other language-related errors.
---

VERSION 1 – AUTHOR RESPONSE

Reviewer: 1

Please state any competing interests or state ‘None declared’:

Authors’ response: Thank you for indicating this. We have now included this statement (Page 16, line 18).

This paper uses a multilevel regression approach to explore distance, difference in altitude, and socioeconomic variables’ impacts on maternal and child healthcare facility utilization in 46 districts of four Ethiopia regions. The topic is of significance and the authors did a fairly good job in describing the research background, strengths and limitations, and methodology. I have several comments before the paper can be accepted for publication.

1. The authors stressed that no previous studies included both distances between household and healthcare facility, and differences in altitude in the same study. It’s best if the authors can provide more details in terms of why including both are important, probably from either the methodological or

theoretical perspective, or both.

Authors' response: Thank you so much for your comment; we have added details on this in the Methods section (Page 8, line 9-11).

2. How did the authors account for the clustering effects? More details are needed.

Authors' response: As individual observations were clustered by geography (region, distance to the health facility and their altitude), we used multi-level regression modelling that considered clustering. We describe this in the methods section (page 8, line 7-9).

3. I had difficulty understanding the characterization of the group-level variables used in the analysis, in particular the distance and the difference in altitude. If the distance/difference is between each household and the regional center or facility locations, why it is group-level rather than individual-level?

Authors' response: Thank you so much for your comment. The selected households are located in the same area – a village - and they are using the same nearest health facility. Thus, the distance to the facility and difference in altitude to that health facility have group-level effects. We have reflected on this in the methods section (page 8, line 7-8.)

4. In Table 3, the authors described the negative association between medium distance stratum and immunization/healthcare utilization as a result of random error. More clarifications are needed given that this finding exists for 3 out of the 4 models.

Authors' response: Thank you so much for your comment. We have modified this statement. We comment that this may be a random error due to the inconsistency across distance tertiles, or, if true, be an effect of where the health post outreach activities take place (page 14, line 16-19)

5. Since the effects of distance to healthcare facilities on adult women and children were different based on Tables 2 and 3, what are the potential policy implications regarding improving women's and children's access to healthcare facilities? Would they be different? I'd like to see more discussions on this.

Authors' response: Thank you so much for the question and suggestion. We have added policy implications in the discussion (page 13, line 6-9.)

6. A related question to Q5 is: while using the straight-line distance between home and the nearest healthcare facility to characterize healthcare access is controversial in the literature, are there any other measures the authors would consider to improve healthcare access characterization in addition to network distance-based measures?

Authors' response: Thank you so much for your comment. Some measures could estimate the actual distance between two points, such as cost time. We were not able to calculate network distance due to data limitations and we have commented upon this in the manuscript, (page 15, line 7-16). We have included the difference in altitude, which reflects the slope between the household and facility.

Reviewer: 2

Please state any competing interests or state 'None declared':

Authors' response: We have now added this under the declaration section (Page 16, line 18).

Abstract

The abstract covers the essential elements of an abstract, however, the following should be considered.

1. Page 3, line 10, replace "socio-economic group" with "socio-economic status"

Authors' response: We have replaced the word socio-economic group with socio-economic status.

2. Page 3, line 14, replace "was" with "were" in the statement "A household and health facility surveys were conducted..."

Authors' response: Thank you. Done.

3. What is the rationale for the 95% confidence intervals around the percentages? This comment applies to the results section as well

Authors' response: Thank you so much for your comment. The confidence interval is used to indicate the range of the true parameter and helpful if comparing our results with other studies.

Introduction

4. Page 6, line 3, replace "stresses" with "stress"

Authors' response: Thank you. Replacement done.

5. Page 6, line 10, replace "socio-economic group" with "socio-economic status"

Authors' response: Thank you so much for your comment; we have replaced the word socio-economic group with socio-economic status.

Methods

6. Page 8 line 31, replace "was using" with "used"

Authors' response: Thank you. Done.

7. Page 8, line 45. The statement "194 enumeration areas were selected with probability proportional to size" is not clear. Please expatiate

Authors' response: Thank you so much for your comment; we have now clarified this.

8. Page 10, line 22-23. Was the nearest facility the same as the facility that was used for the various maternal and child health services under consideration? How was the situation whereby, the nearest facility was bypassed for a distant facility treated in the study?

Authors' response: Thank you for your comment. The nearest health facility differs between maternal health services and child health services. We calculated the distance to the nearest health post for child health services (full child immunization and sick child health care utilization) and the distance to the nearest health center for maternal health services (antenatal care and facility delivery). This has been described in the data management and analysis sub-sections under the methods section (page 7 line 15-17)

9. Page 11, line 33. The statement "patients and the public were not involved in this study" is not clear. From whom was the data collected?

Authors' response: Thank you for your comment; we have corrected in "Patient and/or the public were not involved in the design or conduct or reporting or dissemination plans of this research"

Results

10. Page 12, line 10 "table 2" is hanging. What is about table?

Authors' response: Thank you so much for your comment; Corrected.

Discussion

11. Page 1-13. I am not clear with the connection of the Kenya study on bypassing with the results of the current study.

Authors' response: Thank you so much for your concern; we have included this study because we wanted to show that only distance is not a problem for not using the health facility rather there are some service quality components which could affect the service utilization. We have described this in the discussion section (page 13, line 5-6)

General comment: The entire manuscripts must be edited to correct a number of grammatical and other language-related errors.

Authors' response: Thank you so much for your comment; we have reviewed the whole manuscript and corrected any identified grammar and spelling errors.

VERSION 2 – REVIEW

REVIEWER	Dr. Gilbert Abotisem Abiio University for Development Studies, Ghana
REVIEW RETURNED	01-Dec-2020

GENERAL COMMENTS	The authors have adequately addressed my comments
---